# Current State of Metabolomics Research in Meat Quality Analysis and Authentication

**DOI:** 10.3390/foods10102388

**Published:** 2021-10-09

**Authors:** Tao Zhang, Can Chen, Kaizhou Xie, Jinyu Wang, Zhiming Pan

**Affiliations:** 1College of Animal Science and Technology, Yangzhou University, Yangzhou 225009, China; zhangt@yzu.edu.cn (T.Z.); chencan19981224@163.com (C.C.); kzxie@yzu.edu.cn (K.X.); 2Joint International Research Laboratory of Agriculture and Agri-Product Safety, Ministry of Education, Yangzhou University, Yangzhou 225009, China; zmpan@yzu.edu.cn; 3Jiangsu Key Laboratory of Zoonosis, Key Laboratory of Prevention and Control of Biological Hazard Factors (Animal Origin) for Agrifood Safety and Quality, Ministry of Agriculture and Rural Affairs of the People’s Republic of China, Yangzhou University, Yangzhou 225009, China

**Keywords:** meat quality, meat authentication, metabolomics, nuclear magnetic resonance, mass spectrometry

## Abstract

In the past decades, as an emerging omic, metabolomics has been widely used in meat science research, showing promise in meat quality analysis and meat authentication. This review first provides a brief overview of the concept, analytical techniques, and analysis workflow of metabolomics. Additionally, the metabolomics research in quality analysis and authentication of meat is comprehensively described. Finally, the limitations, challenges, and future trends of metabolomics application in meat quality analysis and meat authentication are critically discussed. We hope to provide valuable insights for further research in meat quality.

## 1. Introduction

Animal meat is an essential part of food that is the primary protein source for the human population [1,2]. In recent years, there has been an increased global demand for meat products. Global meat production has rapidly increased by 25% in the past ten years to 323 Mt in 2017, and it is expected to grow by more than 48 Mt in 2027 [3]. Meat quality can be defined as the set of parameters, attributes, and characteristics that determine the suitability for consumption of fresh or stored meat without any deterioration over a certain time interval [4]. With the opening of the food markets worldwide, meat quality now includes several other aspects such as geographical origin, sophisticated frauds, and adulteration practices [5]. As living standards improve globally, meat quality becomes the critical factor governing consumers’ buying decisions [6]. Recent developments in the meat industry and an increase in public demand for high-quality meat have brought about new challenges, including the efficient assessment of meat quality [4,7]. Therefore, many researchers have devoted themselves to the study of meat quality evaluation and meat authentication. More sensitive, robust, efficient, and cost-effective analytical methods aiming to guarantee the quality of meats are required [8].

Meat quality evaluation is a challenging topic for different analytical techniques [5]. Recently, omics technologies such as genomics, transcriptomics, proteomics, and metabolomics have shown their potential in food compound profiling, food authenticity, and biomarkers analysis related to food quality [9,10]. As an emerging field of omics, metabolomics focuses on comprehensive and simultaneous profiling of the total metabolites in a given organism or biological sample [11]. Currently, metabolomics has been widely used in biomarker discovery, toxicology evaluation, drug research, nutrition research, and crop and farm animal research [12,13,14,15]. Metabolomics has also been successfully applied in various fields of food science [16], showing promise in analysing meat quality and controlling meat safety [17,18]. Here, we provide a brief overview of metabolomics technology, followed by a critical review of the recent advances of metabolomics in the quality assessment and authentication of meat and meat products. Moreover, the challenges, limitations, and future development of metabolomics in meat quality are also discussed. We hope to provide valuable insights for further research in meat quality.

## 2. Metabolomics

### 2.1. Concept of Metabolomics

Horning et al. first reported a metabolic profiles study in 1970 [19]. However, the concepts of metabolomics and metabonomics were defined by Fiehn and Nicholson et al. in 1999 and 2002, respectively [20,21]. Metabolomics is highly related to metabolome. The metabolome is defined as the complete set of small molecules found within biological samples, tissues, and cells [22]. Metabolomics, also called metabonomics, aims at identifying the metabolome, i.e., the complete set of small metabolites (molecular weight < 1500 Da) present in a biological system [23,24]. It is to be noted that two terms, metabolomics and metabonomics, are used synonymously in metabolomic studies. The focus of metabonomics is on understanding systemic change through time in complex multicellular systems. Metabolomics seeks an analytical description of complex biological samples and aims to characterise and quantify all the small molecules in such a sample. In practice, the terms metabonomics and metabolomics are often used interchangeably, and the analytical and modelling procedures are the same [25]. In food science, metabolomics can be described as the application of high-throughput analytic chemistry technologies directed at characterising the food metabolome [26].

Metabolomics can be divided into the categories of untargeted and targeted metabolomics. The untargeted metabolomics approach focuses on the simultaneous detection of many unknown metabolites, while the targeted approach focuses on identifying and quantifying selected metabolites, such as those involved in a particular metabolic pathway [27]. Untargeted metabolomics has the best metabolites coverage; however, it has poor reproducibility and limited sensitivity for low-abundance metabolites. Targeted metabolomics has the advantages of high sensitivity, broad dynamic range, and reliable quantification accuracy, although it covers limited metabolites [28].

### 2.2. Analytical Techniques for Metabonomics

The analytical techniques developed for metabolomics can be defined as particular analytical platforms detecting the set of all metabolites (identified or unknown) in a sample, together with an estimate of the quantity [29]. Nuclear magnetic resonance (NMR) spectroscopy and mass spectrometry (MS) are two predominant analytical techniques used in metabolomics research [30]. NMR has been commonly used in profiling the total complement of metabolites (“fingerprint”) in a sample and is quantitative and does not require extra steps for sample preparation, such as separation or derivatisation. Compared with NMR spectroscopy, MS is superior in allowing analysis of secondary metabolites where the detection level is of picomole to femtomole [31,32]. MS-based metabolomics provides better sensitivity for metabolomics research and wide detection coverage of metabolites [33]. Although NMR and MS are two powerful analytical techniques for metabolomics, detection alone does not always lead to the unambiguous identification of metabolites [21]. The integration of NMR and MS is widely used in metabolomic analysis and achieved greatly improved coverage of metabolites and enhanced accuracy of metabolite identification [34,35,36]. Therefore, the application of the combined NMR and MS approach might become one of the hot topics of metabolomics analysis in the future.

### 2.3. Metabolomic Analysis Workflow of Meat

Commonly used analytical techniques for meat metabolomics include NMR spectroscopy and mass spectroscopy. MS is often used in combination with gas chromatography (GC-MS) or liquid chromatography (LC-MS). The workflow of metabolomics analysis generally consists of three steps: sample preparation, metabolomic analysis, and data interpretation [37] (Figure 1). In the present review, we will focus on the sample preparation and data interpretation steps.

#### 2.3.1. Sample Preparation

It is essential to choose a suitable sample preparation method in metabolomics research because it affects both the metabolite identification and interpretation of the data [38]. An ideal sample preparation protocol should: (i) be as simple as possible to ensure its reproducibility; (ii) be fast to prevent metabolite degradation during the preparation procedure; and (iii) be low cost [39]. In general, meat samples are prepared in a two-step process of sample collection and metabolite extraction. There is no significant difference in the sample collection of NMR spectroscopy, GC-MS-, and LC-MS-based metabolomics analysis. Collecting samples requires special care because rapid metabolite changes caused by enzymatic degradation or microbial activity could occur during the process and may affect the results considerably [40]. For this reason, the collecting of the samples should be conducted rapidly at low temperatures. Moreover, the collected samples should be quenched by liquid nitrogen immediately to guarantee true metabolome composition at the sampling time. For quenched samples, metabolite extraction should be performed rapidly. Otherwise, samples must be stored at a low temperature (−80 °C or lower temperatures are recommended).

The metabolite extraction step follows sample collection. Meat samples should be homogenised before the extraction process. Yield, reproducibility, ease, and speed are the standards for evaluating the quality of an extraction method for metabolomics [41]. For NMR, samples are prepared using destructive and non-destructive methods. The non-destructive method does not require metabolite extraction and is commonly used for NMR-based metabolomics that detects metabolites of intact tissue in situ by using the magic angle spinning (MAS) technique. However, the destructive sample preparation method is recommended in meat NMR-based metabolomics studies. The destructive method needs to have an extraction procedure performed before NMR detection. The principles of metabolite extraction should be as follows: (i) before detection by NMR, the enzyme activity should be terminated (which can be achieved by utilising acid or organic solvents such as methanol, ethanol, or acetonitrile) and (ii) the maximum amount of metabolites in meat samples should be extracted with appropriate methods [41,42]. Usually, the metabolite extraction of meat samples is performed with the use of solvents such as deuterium oxide [43], methanol [44], perchloric acid [45], phosphate buffer [46], chloroform [47], and their combinations. Although methanol is the most commonly used solvent for metabolite extraction of meat samples, recent studies have indicated that the methanol-chloroform combination seems to be an optimal solvent considering both yield and reproducibility [41,48].

The strength of GC-MS is the measurement of non-polar and volatile organic compounds [49]. Therefore, extraction that maximises the number and amounts of metabolites combined with derivatisation that transforms polar compounds into volatile compounds is necessary before GC-MS analysis [32]. Methanol, chloroform, and combination are the most applied solvents in metabolite extraction of meat samples [50,51]. The derivatisation can be achieved by the trimethylsilylation derivatisation reaction on thoroughly dried samples at room temperature with pyridine as the catalyst [52]. Recently, solid-phase microextraction (SPME) has been widely used in food science and has been proven to be an ideal method to extract metabolites from meat matrices due to its simple and solvent-free characteristics [53,54,55].

For the metabolome to be analysed by LC-MS, it must be placed in solution. Thus, a homogenisation and extraction step is essential for meat samples to solubilise metabolites. Ceramic or metal beads and orbital shaking are primarily used for homogenisation. The most widely used metabolite extraction methods for LC-MS analysis include (i) organic solvent extraction; (ii) liquid-liquid extraction; and (iii) molecular cut-off weight filters. The choice of these methods depends on the presence of macromolecules that can damage the LC-MS system, polarity, and concentration of the metabolites found in meat samples. The organic solvent extraction method is the most general procedure for all biospecimens for its versatility and simplicity. In this approach, methanol, acetonitrile, isopropanol, or their mixtures are commonly used organic solvents [56,57]. Acetonitrile or methanol, plus water solutions, is suitable for extracting polar metabolites [58,59], while isopropanol or LLE is suitable for lipids [60,61].

#### 2.3.2. Data Interpretation

Measuring metabolites and interpreting their biological relevance within the contexts of different experimental conditions are the primary objectives in metabolomics research [62]. Data interpretation of metabolomics data relies on two steps: data preprocessing and pretreatment and biological interpretation.

##### Data Preprocessing and Pretreatment

Efficient and reliable data preprocessing is the first step towards successful data analysis and biologically important findings. In general, preprocessing of NMR metabolomics data involves apodisation, Fourier transform, phasing, baseline correction, and chemical shift calibration [63]. After preprocessing, NMR data are transferred to an NMR spectrum data matrix consisting of chemical shift and peak intensity information [64]. Unlike NMR spectroscopy, LC/GC-MS analysis generates data files consisting of a complex three-dimensional (3D) data format comprising retention time, m/z values, and density or abundance on each axis [65]. The preprocessing aims to transform the 3D data table into a 2D format with the rows corresponding to samples and the columns to m/z-RT pairs through peak picking/detection and deconvolution, alignment and gap filling, and quality control. Several tools are available to perform the initial preprocessing steps, such as Mzmine [66], OpenMS [67], XCMS [68], and apLCMS [69]. Quality control is conducted by adding quality control samples (QCs) after every couple of (between 5 and 10) study samples in the entire sample run [70].

A normalisation step following the preprocessing is needed to remove unwanted variation between the samples and allow quantitative comparison of the samples [63]. The normalisation can be performed in several ways, including the addition of internal/external standards, total area normalisation, probabilistic quotient normalisation, and quantile normalisation [71,72,73,74]. The preprocessing and normalisation generate clean and normalised metabolomics data that are ready for subsequent analysis. However, an appropriate data pretreatment step is necessary to reduce the effects of technical and measurement errors before starting [75]. It is to be noted that data pretreatment is generally needed when multivariate analysis methods are considered. There are mainly two methods for data pretreatment: centring and scaling and data transformation. Centring adjusts for differences in the offset between high and low abundant metabolites, and scaling adjusts for the fold differences between the different metabolites [75]. Centring and scaling of metabolomics data can be accomplished by auto-scaling, Pareto scaling, range scaling, and vast (variable stability) scaling operations [76,77]. Transformations are nonlinear data conversions by log transformation, glog transformation, or power transformation, generally aiming to correct heteroscedasticity [78].

##### Biological Interpretation

Biological interpretation is one of the critical steps in metabolomics study. However, it is becoming increasingly challenging to efficiently interpret changes in metabolite levels and determine their biological significance due to the growing metabolomics datasets. A large group of statistical methods and software has been developed to address this issue. Herein, we focus on describing multivariate techniques for the subsequent analysis of metabolomics data, including principal component analysis (PCA), partial least squares discriminant analysis (PLS-DA), and orthogonal partial least squares discriminant analysis (OPLS-DA). Machine learning (ML) methods and functional analysis of metabolites are also outlined.

The oldest and most widely used multivariate technique in metabolomics is PCA [79]. PCA is a powerful means of analysing metabolomic data and is usually used as the first step in the analysis of metabolomics data [80]. Conversion of the original dataset by PCA results in two matrices known as scores and loadings. PCA provides an overview of all samples in the data table by inspecting the relationship between scores and loadings. In addition, groupings, trends, and outliers in the sample can also be detected [81].

In many metabolomics research studies, the interest lies in discriminating two or more groups to select variables (i.e., metabolites) that are important to the studied biological problem. This is primarily conducted in a multivariate context using discriminating techniques, such as partial least squares discriminant analysis (PLS-DA) [82]. Unlike PCA, PLS-DA is a supervised method extending from PLS [83]. This approach aims to maximise the covariance between the independent variables X (metabolomics data) and the corresponding dependent variable Y (classes, groups) of highly multidimensional data by finding a linear subspace of the explanatory variables [84]. PLS-DA holds many advantages over PCA. However, PLS-DA tends to construct overly complex models when processing metabolomics data [85]. For this reason, OPLS-DA is usually used instead of PLS-DA to construct more parsimonious and easily interpretable models by disentangling group-predictive and group-unrelated variation in the measured data [86]. OPLS-DA is a modification of the PLS-DA method and provides better performance in separating predictive from non-predictive (orthogonal) variation [85]. Numerous studies have shown the potentialities and applications of PCA, PLS-DA, and OPLS-DA in handling meat metabolomics data [43,55,87,88,89,90].

Many online tools and software packages are available for multivariate analysis of metabolomics data: MetaboAnalyst [91], MVAPACK [92], SAS (SAS Institute Inc., Cary, NC, USA), the PLS toolbox for Matlab (Eigenvector Research Inc., Wenatchee, WA, USA), and SIMCA (Umetrics, Umeå, Sweden). In comparison to the other software packages, SIMCA is much more widely used in the metabolomics field. The detailed introduction of SIMCA was described by Triba et al. [93]. Notably, the performance of PLS-DA/OPLS-DA models should be estimated by cross-validation or permutation test because PLS-DA/OPLS-DA models can easily be overfitted and their predictability overestimated. For PLS-DA and OPLS-DA models, variable importance in projection (VIP) value is used to estimate the importance of each variable and select biomarkers.

Metabolomics data analysis includes two main types: regression and classification. PLS-DA and OPLS-DA are commonly used to construct classification models. Regression analysis is needed when the responses attached to each sample are continuous. For regression analysis, multiple linear regression (MLR) is one of the basic models [94]. OPLS regression (OPLS-R) is another model used frequently in metabolomics research. Both MLR and OPLS-R models have shown their potentiality and ability in identifying meat-quality-related biomarkers [56,95].

Recently, the increasing big data set generated by large-scale studies on the metabolome poses a new challenge for metabolomics research. Machine learning (ML) methods have become immensely popular for statistical analysis of metabolomics data due to their ability to rapidly process large and heterogeneous data [96]. Machine learning can be described as a set of algorithms that improve prediction accuracy through experience, given a certain processable input from which they can learn and generalise [97]. Support vector machines (SVM) and random forests (RF) are the two most used and powerful ML algorithms applied to metabolomics study. SVM is an effective non-parametric machine learning algorithm suitable for both classification and regression problems. This algorithm is based on mapping data into a high-dimensional space that allows for separating two groups of samples into distinctive regions. Compared to PLS-DA and OPLS-DA, SVM is not affected by the distribution of the different sample classes [98]. The main advantage of the SVM algorithm is its flexibility in choosing the kernel function that allows the separation of two groups of samples, and this kernel can be chosen for either linear or nonlinear problems [99]. A significant drawback of SVM is its restrictions on binary classification problems. For example, it can only discriminate between two classes where the data points are categorised by two classes in n-dimensional space, where n corresponds to the number of metabolites [100]. RF belongs to the family of classification trees and is found to be the best classifier [101,102]. RF has the strength to deal with large datasets without variable deletion, to provide a feature importance measure of the metabolite (mean decrease in accuracy) and a measure of the internal structure of the data (mean decrease in Gini index), and to handle missing values [84].

##### Functional Analysis

Metabolomics aims at characterising the profiles of metabolites in a biological sample. As more massive and larger sets of metabolites are detected, a functional analysis is required to convert these raw lists of metabolites into biological knowledge [103]. Perhaps the most considerable challenge that metabolomic researchers face in any study is relating the identified metabolites to their biological roles [13]. The most common method of performing such an analysis is “functional enrichment analysis”. The functional analysis requires a knowledge database defining functionally related molecule groups and a statistical algorithm to perform enrichment tests [63]. In metabolomics, except the public metabolic pathway databases KEGG [104], metabolite set enrichment analysis (MSEA) and ConceptMetab database provide the comprehensive metabolite annotation based on GO, KEGG pathway, and human disease [105,106]. Recently, an easy-to-use web-based tool, MetaboAnalyst, was developed to perform comprehensive metabolomic data analysis, interpretation, and integration. This tool integrated various functions such as PCA, PLS-DA, clustering analysis and visualisation, MSEA, metabolic pathway analysis (MetPA), biomarker identification, and time series and power analysis [91]. MetaboAnalyst has recently been updated to the current version 5.0 and numerous studies have shown its ability to analyse metabolomics data. The above database and tools are powerful when dealing with metabolomics data obtained from human or rodent. However, their applications in handling metabolomics data of meat are relatively scarce due to the limited annotation information of metabolite in meat samples.

## 3. Metabolomics in Meat Quality and Authentication

Meat and meat products are highly appreciated due to their sensory properties and nutritional composition [107]. As a global issue, food safety and quality receive increasing attention from both businesses and customers. People currently pay more attention to the quality and authenticity of meat [108]. Thus, efficient methods are needed to assess the quality and authenticity of meat. As an emerging analytical platform, metabolomics has been widely applied to evaluate meat’s freshness, composition, authenticity, and originality (Figure 2, Table 1).

### 3.1. Metabolomics in Meat Quality

Meat quality has always been important to the consumer, and it is an especially critical issue for the meat industry in the 21st century. Generally, meat quality can be divided into appearance quality traits (AQT), eating quality traits (EQT), and reliance quality traits (RQT) [109]. Here, we mainly focus on reviewing the metabolomics research in colour, pH, and meat texture of AQT, tenderness and flavour of EQT, and freshness of RQT.

#### 3.1.1. Appearance Quality Traits (AQT)

Meat colour is the most important AQT because it is the first factor seen by the consumer. The colour of meat, especially beef and mutton, is an important deciding factor in consumers’ assessment of meat quality. The bright red colour is usually seen as an indicator of freshness and overall wholesomeness of meat [110]. However, a loss in meat colour usually occurs during storage, accompanied by changes in pH, creating a constant demand by retailers for assurance on the colour and colour stability of the meat supplied [111]. Therefore, understanding the chemical basis of discolouration in meat is required to develop methods to maintain meat’s acceptable colour stability. Dark-cutting beef is an example of a colour discolouration in which beef fails to have a characteristic bright red colour. Dark cutting is produced worldwide, leading to significant economic losses to the food industry [112]. Recent studies sought to investigate the biochemical basis for the development of dark-cutting beef using GC-MS- and LC-MS/MS-based nontargeted metabolomics approaches. The authors found that changes in pH and colour of dark-cutting beef were probably caused by the upregulated tricarboxylic metabolites and increased mitochondrial content, and downregulated glycolytic metabolites and glycogen degradation enzymes [113,114]. Beef colour is affected by the interrelationship between mitochondria and myoglobin function. Other researchers have come to similar conclusions that increased mitochondrial damage, depletion of metabolites that can regenerate NADH, and increased oxidative stress decrease colour stability in aged beef [115]. Many metabolite responses to the ageing of beef were also identified, such as acyl carnitines, free amino acids, nucleotides, nucleosides, and glucuronides [147]. For ovine meat, the colour stability might be associated with myoglobin chemistry and antioxidant-activity-related metabolites [111].

Meat pH influences the paleness of the raw meat, toughness after cooking, and water-holding capacity during storage and processing [120]. The processing ability and sensorial quality of poultry meat are determined by meat’s ultimate pH (pHu). Beauclercq et al. attempted to identify biomarkers to predict ultimate pH by detecting discriminating metabolites in the muscle and serum between the pHu− and the pHu+ chicken lines using an NMR-based metabolomics method. It was found that chickens in the pHu− line used carbohydrates as the primary energy source, whereas those in the pHu+ line used energy produced from amino acid catabolism and lipid oxidation. Several discriminating metabolic markers that could be used to predict pHu were highlighted, including glucose, betaine, taurine–betaine, dimethylglycine, arginine–lysine, and mannose 6-phosphate for muscle, 3-methylhistidine, xanthine, 1-methylhistidine, glucose, arginine, glutamine, and maltose for serum [148]. Furthermore, the slow pHu drop in pork was proven to be related to higher levels of glycolytic enzymes and lactate accumulation [116]. In sheep meat, a significant correlation was found between 1, 5-anhydroglucitol and meat pH [117].

Meat texture is another AQT that is partially affected by the quantity of intramuscular fat (IMF) [109]. In animal production, intramuscular fat (IMF) is positively related to meat quality, including tenderness, flavour, and juiciness. Thus, understanding the cell origin and regulation mechanism of IMF infiltration is important for improving meat quality [118]. Metabolomics can be used as a complementary tool of proteomics and transcriptomics to address this issue. Liu et al. performed a combined metabolomics and transcriptomics approach to explore the effect and regulation mechanism of CRTC3 on porcine intramuscular adipocyte differentiation [119]. This study revealed that CRTC3 regulates glycerophospholipid metabolism-related genes and promotes increased phospholipid formation to enlarge adipocytes for more lipid storage. In addition, CRTC3 promotes IMF deposition by upregulating the Ca^2+^-cAMP signalling pathway and downregulating fatty acid metabolism capacity in intramuscular adipocytes. The IMF’s phenotypic and genetic selection is difficult for the breeders, as it can only be accurately measured after slaughter [149]. An alternative way to address this challenge is identifying plasma biomarkers related to IMF content in meat. With the applications of metabolomics, numerous blood or plasma biomarkers correlated with IMF content in bovine and porcine meat have been identified, including branched-chain amino acids, 3-hydroxybutyrate, propionate, acetate, creatine, histidine, valine, and isoleucine [121,150]. These biomarkers could help understand IMF deposition better and predict the IMF trait in situ instead of post slaughter.

#### 3.1.2. Eating Quality Traits (EQT)

Tenderness is the most important EQT because it strongly influences consumers’ perceptions of acceptability. In the past decades, standard investigations of meat tenderness have mainly relied on physical and mechanical measurements through tests for cut resistance, meat colour, and pH value [151]. More recently, the technical advancements in NMR- and MS-based metabolomics provided powerful new strategies to delve into the tenderness issue. D’Alessandro et al. assessed whether metabolites could be predictors of beef tenderness using LC-MS-based metabolomics. The authors indicated that higher levels of glycolytic enzymes characterised tender meat, and the metabolite oxidised glutathione (GSSG) could be considered a biomarker for predicting meat tenderness [122,123].

In addition to the tenderness, the flavour is another main attribute determining consumers’ decisions to purchase meat [152]. Meat flavour is a combination of taste and odour. Non-volatile constituents of fresh meat (sugars, peptides, amino acids, inorganic salts, and organic acids) and flavour enhancers (inosine 5′-monophosphate, guanosine 5′-monophosphate, and monosodium glutamate) have been confirmed to be the flavour precursors contributing to the basic tastes of cooked meat [153,154]. However, there are still a large number of flavour precursors contributing to meat sensory characteristics that need to be identified because meat contains potentially hundreds of components that influence its flavour and taste characteristics [129]. Metabolomics has emerged as a powerful tool to estimate flavour precursors in meat. For example, researchers clarified the key metabolites contributing to the rich and sweet aroma of Wagyu beef using a GC-MS-based metabolomics approach and confirmed that the amounts of odorants were highly correlated with glutamine, decanoic acid, lactic acid, and phosphoric acid [124]. Similarly, NMR- and LC-MS-based metabolomics methods were applied to evaluate the chemical composition of precursor flavour in chicken meat. Aroma compounds such as thiazole, 2-methyl-3-furanthiol, 2-furfurylthiol, dihydro-2-methyl-3(2H)-thiophenone, 2-acetylthiazole, and pyrazine were identified as potential contributors to the overall sensory quality of cooked meat [125]. Alanine, aspartate, and glutamate metabolism, purine metabolism, glycine, serine, and threonine metabolism, and taurine and hypotaurine metabolism were demonstrated to be the primary metabolic pathways for the chicken meat flavour [126]. Metabolomics was also used to evaluate the taste attributes of fish meat. Strong associations were found between “sourness” and lysine, “irritant” and alanine and phenylalanine, “saltiness” and pantothenic acid, and “umami” and creatinine and histidine in fish meat [127]. Phosphoric acid was identified as a candidate marker for evaluating differences in the taste of four fish species: *T. modestus*, *I. japonicus*, *S. marmoratus*, and *P. major* [128].

Maillard’s reaction produces volatile flavour components responsible for the characteristic cooked meat aroma [153]. With gas chromatography coupled with time-of-flight mass spectrometry (GC–TOF/MS), the relationship between volatile compounds and the sensory attributes of glutathione-Maillard reaction products (GSH-MRPs) in beef was investigated, in which volatile compounds such as 2-methylfuran-3-thiol, 3-sulfanylpentan-2-one, furan-2-ylmethanethiol, 2-propylpyrazine, and 1-furan-2-ylpropan-2-one could be identified as possible critical contributors to the beef-related attributes of GSH-MRPs [129].

Dry-cured ham is a popular cured meat product with high storage stability and typical sensory characteristics [155]. Metabolomics has been used to examine the chemical changes in dry-cured ham during the ripening process, aiming to identify key chemical components for characterising the taste and flavour of ham. For example, metabolomic profiles of dry-cured ham during the ripening process were characterised by NMR-, CE-MS-, and GC-MS-based metabolomics approaches. Amino acids, organic acids, and nucleotide derivatives were major contributors to the taste of boneless and Japanese Prosciutto dry-cured hams. The taste of dry-cured ham was significantly affected by the processing time [130,131]. Another research study used a nontargeted metabolomics approach to characterise volatile flavour compounds in the Dahe black pig ham. Hexanal, 3-methyl-butanal, nonanal, and octanal were identified as characteristic flavour components [132]. Moreover, recent studies also showed excellent performance of metabolomics in analysing the metabolic differences in different dry-cured hams, characterising taste substances of ham with different processing procedures and conditions [133,134,135].

#### 3.1.3. Reliance Quality Traits (RQT)

Freshness is identified as one of the most critical RQTs in evaluating the quality and safety of meat [7]. The freshness of the meat is negatively correlated to spoilage caused by a variety of microbial activities. The spoilage process caused by microbial activity produces a large number of low-molecular-weight metabolites. The analysis and characterisation of these metabolites can provide crucial information for meat control, classification, and quality assessment [56]. Metabolomics has been applied in characterising these metabolites and measuring the freshness of meat. For example, Zhang et al. and Wen et al. developed UHPLC-MS-based untargeted metabolomics to measure the freshness of chicken, and multiple freshness-related metabolic biomarkers were identified, such as L-anserine, tyramine, and indole-3-carboxaldehyde [56,57]. Beef is one of the meat products with increased demand and commercial value [90]. Recent studies implied that NMR- or GC-MS-based metabolomics could classify beef samples according to their freshness and predict the storage time of beef samples. Additionally, numerous potential metabolic biomarkers related to the freshness of beef were identified, including 2-pentanone, 2-nonanone, 2-methyl-1-butanol, 3-methyl-1-butanol, ethyl hexanoate, ethyl propanoate, ethyl lactate, ethyl acetate, ethanol, 2-heptanone, 3-octanone, diacetyl, and acetoin [87,156]. For pork, a study was carried out to monitor the metabolic changes during storage, which laid a foundation for developing a new method for non-destructive analysis and for the control of pork quality [136]. In another study, GC-MS-based metabolomics was applied to metabolic changes of Tan sheep meat during storage. Gluconic acid, citric acid, trans-4-hydroxy-L-proline, and 1, 5-anhydroglucitol were identified as potential spoilage biomarkers of sheep meat [117]. Compared with livestock and poultry meat, metabolomics is more widely used in evaluating freshness and identifying freshness-related biomarkers of fish meat such as yellowtail [51], tuna [157], gilthead sea bream [137], tilapia [158], and komatsuna [159]. The above studies demonstrate the feasibility of metabolomics in estimating meat freshness, especially identifying freshness-related biomarkers.

### 3.2. Metabolomics in Meat Authentication

Adulteration of foods is a severe economic problem concerning most foodstuffs, particularly meat and meat products [160]. Since adulteration can have severe consequences on human health, it affects market growth by destroying consumer confidence [161]. Therefore, authentication of meat is essential to ensure fair competition, consumer benefit, and food safety. Adulteration of meats can be divided into four categories, including meat origin, the replacement of higher quality meats with lower quality ones, the substitution of meat muscle proteins with vegetable proteins such as soybean, and the addition of unsaid meat species and unsaid ingredients in meat-based food products [162]. Recently, there is a growing need for new analytical methods to guarantee that all the ingredients included in a foodstuff match the qualities claimed by the manufacturer [163]. Nevertheless, it is usually challenging to differentiate between adulterated and pure meat using conventional sensory techniques and quality indicators [16]. With the development of food omics, metabolomics has emerged as a powerful approach in assessing meat’s authenticity by characterising its chemical composition and metabolite contents.

#### 3.2.1. Geographical Origin

Meat products, particularly protected geographical indication (PGI) products of selected breeds produced in a particular area, have a higher value in the market [90,164]. With improved living standards, consumers are becoming increasingly aware of the importance of meat origin, such as geographical origin and species origin [138]. Consequently, there is a growing need to develop appropriate analytical methods to determine meat’s geographical origin and species origin. The discrimination of geographical and species origins to guard the protected designation of origin (PDO) and avoid fraudulent labelling of meat products has been intensively studied using NMR- and MS-based metabolomics techniques.

The geographical origin of beef is of increasing interest to consumers and producers due to “mad cow” disease and the implementation of the Free Trade Agreement (FTA). An NMR-based metabolomics method was used to discriminate beef originating from four countries: Australia, Korea, New Zealand, and the United States [90]. Primary metabolites responsible for discrimination of the geographical origin of raw beef were identified, including succinate and various amino acids. In another study carried out by MS-based metabolomics, Man et al. characterised the metabolite profiles of beef samples from different geographical origins [139]. Twenty-four metabolites were identified as metabolic biomarkers for beef from different countries, including amino acids, several sugar metabolites, and many PCs and PEs. Regarding lamb meat, the metabolomics approach based on stable isotope ratios and NMR achieved 100% of classification ability and 96% of prediction ability in classifying lamb types according to their geographical origins [140]. Likewise, the metabolomics approach has been applied to discriminate the geographical origin of aquatic products such as Mytilus [141] and shrimp [165]. These studies demonstrate that metabolomics is an efficient method to discriminate the geographical origin of meat.

#### 3.2.2. Species Origin

The high value of meat opens it up to fraudulent replacement/substitution of some or all of the premium meat content with lower-grade meat or meat from other species [142]. Thus, there is a need to develop efficient and high-throughput analytical approaches to detect meat adulteration. Beef is more expensive compared with other conventional types of meat such as chicken, pork, or horse meat. Substitution of the more expensive beef with cheaper pork or chicken is an attractive proposition for those inclined to adulterate the food supply for economic gain. Several recent studies have exploited NMR- and MS-based metabolomics as methods to detect the adulteration of beef with pork [55,143]. Both studies confirmed metabolomics’ potential as an alternative method for robust and reliable discrimination of adulterated and pure beef samples by constructing the OPLS-DA or PLS-DA model. Many metabolites that correlated with beef, pork, and their mix were identified, respectively. Akhtar et al. performed a ^1^H-NMR-based metabolomics study to investigate the metabolic difference between chicken, chevon, beef, and donkey meat. Lactate, creatine, choline, acetate, leucine, isoleucine, valine, formate, carnitine, glutamate, 3-hydroxybutyrate, and α-mannose were found as the significant discriminating metabolites between white (chicken) and red meat (chevon, beef, and donkey). While inosine, lactate, uracil, carnosine, format, pyruvate, carnitine, creatine, and acetate were found responsible for differentiating chevon, beef, and donkey meat [166]. Moreover, Jakers et al. developed a 60 MHz ^1^H-NMR-based targeted metabolomics to differentiate between beef and horse meat and concluded that 60 MHz ^1^H NMR represents a feasible high-throughput approach for screening raw meat [167]. Lamb meat is derived from sheep at the age of no more than 12 months or without permanent incisor teeth, and mutton is defined as meat from sheep of 1–3 years old. Compared with mutton, lamb meat always has a higher demand and retail price in the market [144]. In order to pursue economic interests, illegal producers often use mutton to replace lamb meat. The high-throughput metabolomics approach combined with multivariate data analysis has been proven to distinguish lamb from mutton effectively [145].

The breed is one of the important factors that affect meat quality. Generally, most native livestock and poultry possess high-quality meat characterised by unique flavours and tastes and high economic value compared to commercial lines. Therefore, it is necessary to develop efficient methods to distinguish meat with different breed origins. For this purpose, metabolomics-based approaches were developed and have been applied to discriminate pork [146], chicken [45,168], and duck [43] meat of different breeds. By the NMR-based metabolomics method, the amino acid carnosine was identified as the metabolite most strongly correlated to the sensory attributes of the pork meat from different breeds [146]. Kim et al. developed a combined ^1^D ^1^H NMR and ^2^D HSQC NMR approach to quantify metabolites present in chicken breast meat extracts from Korean native chickens and commercial broilers [45]. Compared with commercial broilers, Korean native chicken meat possesses higher amounts of IMP, α-glucose, lactate, and anserine and lower amounts of free amino acids. A further study reported by Kim et al. developed a metabolomics approach based on 2D HSQC analysis to differentiate between Korean native chickens and white-semi broiler, demonstrating superior performance to the conventional quality assessment tools [168].

## 4. Challenge and Future Trends

Thanks to its high sensitivity, high throughput, and reliability features, metabolomics has emerged as a powerful tool for analysing meat quality. However, NMR- and MS-based metabolomics applications for meat quality and authentication are still far from reaching their maximum potential. Several technical challenges regarding the application of metabolomics remain, including (1) the complexity of meat samples, (2) the difference in metabolites caused by using different sample preparation methods and instrument platforms, (3) the absence of a specialised database for meat metabolome, (4) the lack of uniform criteria for metabolite identification, (5) the limited information available for functional annotation of metabolites, and (6) the growing dataset generated from large samples [169]. Therefore, to obtain a deeper comprehension of meat metabolome and identify metabolites used for meat quality analysis and authentication, it is necessary to develop harmonised and normalised sampling methods, establish a food metabolome database with functional annotations, develop uniform criteria for metabolite identification, and develop novel powerful exploiting tools [16,169,170].

For decades, remarkable progress has been achieved in meat science, benefiting from improved NMR- and MS-based metabolomics. However, meat is a complex matrix consisting of enormous organic compounds, and its quality is affected by many factors. Still, there exists a large portion of metabolites related to meat quality waiting to be identified. It is necessary to develop methods with higher resolution, higher sensitivity, and better quantitative capability for investigating meat metabolome and identifying biomarkers related to meat quality and authentication. In this context, selected reaction monitoring (SRM) will find more applications in monitoring concentration changes of endogenous metabolites in the targeted analysis of meat. At the same time, full-scan high-resolution mass spectrometers (HRMS) with high mass resolution and high mass accuracy hold the promise of untargeted metabolomics analysis of meat in the future [171]. Notably, although the improvement of analytical techniques of metabolomics makes it possible to analyse hundreds of metabolites in a single run, identifying and characterising the detected metabolites is a challenge posed to the researchers. Thus, developing methods for more efficient identification of unknown compounds and establishing databases for meat metabolome are other issues to be addressed in the meat metabolomics community.

Due to innovative developments in informatics and analytical technologies, metabolomics’ power has extended from biomarker discovery to understanding the mechanisms underlying phenotypes [13]. Nevertheless, most of the metabolomics studies in meat science focus on characterising metabolic profiles, identifying biomarkers, and discriminative analyses. The functions of metabolites in the formation of meat quality and the mechanisms behind them remain largely unknown. How to correlate metabolites to meat quality, investigate the functions of metabolites, and elucidate the mechanisms underlying the function is another challenge. Recently, metabolomics coupled with genomics, transcriptomics, and proteomics has been applied to investigate food and nutrition domains, providing fast, accurate, and reliable tools to address problems inherent to food quality control. Metabolomics’ association with other analytical techniques such as transcriptomics and genomics could be powerful strategies for meat quality analysis because metabolomics can complement other omics methods to provide correlations between metabolic changes and phenotype of meat, offering a more holistic molecular perspective to study meat science comprehensively. Nevertheless, the effective integration of multi-omics data remains challenging, requiring co-progress of systems biology and computer technology.

Meat metabolomics is expected to become a potent tool in quality analysis and authentication to comprehensively characterise the complex meat matrices. Although many successful research projects have already demonstrated the feasibility of metabolomics approaches in characterising metabolomics profiles and identifying biomarkers, their uptake and implementation into routine analysis and meat surveillance are still limited [172]. This is mainly because many studies were performed within a limited period of time using one instrument within one laboratory, limiting the applicability of the developed metabolomics approaches in the meat industry and meat processing. Moreover, there is a lack of validation strategies that guarantee the metabolomics data’s reliability and allow conclusions on the applicability of the metabolomics approaches in meat quality analysis and authentication. Therefore, future works should focus on developing generic schemes to validate the metabolomics-based analytical method in meat quality control and authentication. Furthermore, up-to-date, extensive metabolomics research studies have identified a considerable number of biomarkers related to meat quality and authentication. Another issue that needs to be addressed is the development of targeted metabolomics approaches and other analytical methods such as sensory evaluation based on the identified biomarkers.

## Figures and Tables

**Figure 1 foods-10-02388-f001:**
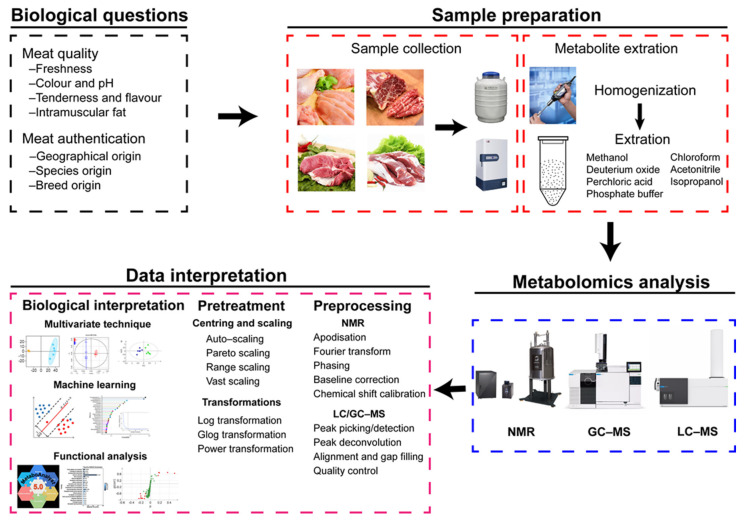
Metabolomic analysis workflow of meat sample.

**Figure 2 foods-10-02388-f002:**
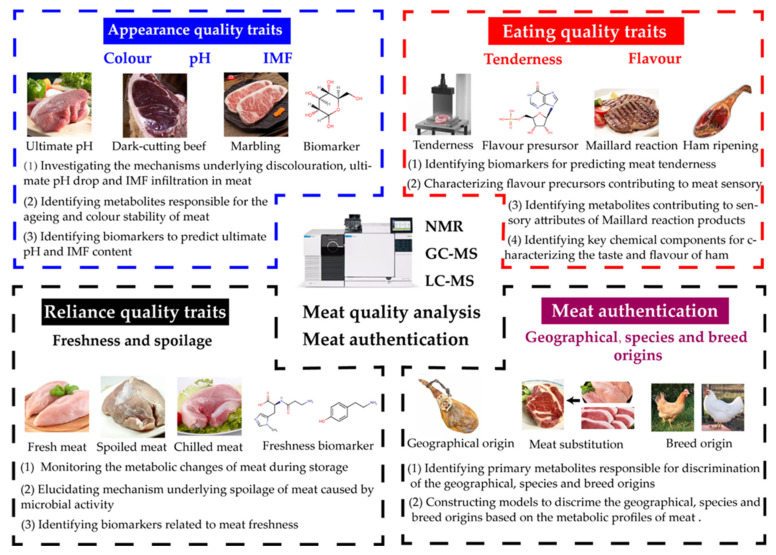
The application of metabolomics in the research into meat quality analysis and authentication.

**Table 1 foods-10-02388-t001:** Summary of recent applications of metabolomics in meat quality analysis and authentication.

Purpose of Study	Meat Type/Species	Analytical Techniques	References	Authors
Meat freshness	Chicken	UHPLC-MS	[56]	Zhang et al.
	Chicken	UHPLC-MS	[57]	Wen et al.
	Beef	NMR	[87]	Castejón et al.
	Beef	GC-MS	[109]	Argyri et al.
	Pork	NMR	[110]	García-García et al.
	Sheep	GC-MS	[111]	You et al.
	Yellowtail	GC-MS	[51]	Mabuchi et al.
	Tuna	UPLC-HRMS	[112]	Chang et al.
	Gilthead sea bream	GC-MS	[113]	Mallouchos et al.
	Tilapia	NMR	[114]	Zhao et al.
	Komatsuna	NMR	[115]	Li et al.
Colour and pH	Beef	GC-MS	[116]	Ramanathan et al.
	Beef	GC-MS	[117]	Kiyimba et al.
	Beef	GC-MS	[118]	Mitacek et al.
	Beef	HPLC-ESI-MS	[119]	Ma et al.
	Mutton	LC-MS	[120]	Subbaraj et al.
	Chicken	NMR	[121]	Beauclercq et al.
	Mutton	GC-MS	[111]	You et al.
Tenderness and flavour	Beef	LC–ESI–CID/ETD–MS	[122]	D’Alessandro et al.
	Beef	LC–ESI–CID/ETD–MS	[123]	D’Alessandro et al.
	Beef	GC-MS	[124]	Ueda et al.
	Chicken	LC-MS	[125]	Zhou et al.
	Chicken	NMR	[126]	Xiao et al.
	Yellowtail	GC-MS	[127]	Mabuchi et al.
	T. modestus, I. japonicus, S. marmoratus and P. major	GC-MS	[128]	Mabuchi et al.
	Beef	GC-TOF/MS	[129]	Lee et al.
	Ham	NMR	[130]	Zhang et al.
	Ham	CE-MS	[131]	Sugimoto et al.
	Ham	GC-MS	[132]	Shi et al.
	Ham	NMR	[133]	Zhang et al.
	Ham	NMR	[134]	Zhou et al.
	Ham	CE-TOFMS	[135]	Sugimoto et al.
Intramuscular fat	Pig	LC-MS	[136]	Liu et al.
	Pig	CE-TOF/MS	[137]	Taniguchi et al.
	Cattle	NMR	[137]	Connolly et al.
Geographical origin	Beef	NMR	[90]	Jung et al.
	Beef	GC-MS	[138]	Man et al.
	Lamb meat	NMR	[139]	Sacco et al.
	Mytilus	NMR	[140]	Aru et al.
	Shrimp	LC-HRMS	[141]	Chatterjee et al.
Species origin	Beef and pork	GC-MS	[55]	Pavlidis et al.
	Beef and pork	GC-MS/UHPLC-MS	[142]	Trivedi et al.
	Chevon, beef, and donkey	NMR	[143]	Akhtar et al.
	Mutton and lamb meat	UHPLC-QTOF	[144]	Wang et al.
Breed origin	Pork	NMR	[145]	Straadt et al.
	Chicken	NMR	[45]	Kim et al.
	Chicken	NMR	[146]	Kim et al.
	Duck	NMR	[43]	Wang et al.

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
