# Peer review of "Current State of Metabolomics Research in Meat Quality Analysis and Authentication"

_foods, 2021, doi:10.3390/foods10102388_

Round 1

Reviewer 1 Report

 This review paper about metabolomics technology for meat is well organized and written. Considering the recent development of analysis technology and the growing importance of evaluating or predicting meat quality andf using these technologies to secure the quality and trust of meat, I think this review paper has a timely meaning. However, one thing I would like to advise for this review paper is that the authors suggested four traits of meat quality decisins, and please refer to Joo et al. (2013, Meat Science 95, 828-836) paper, which divides meat quality into AQT (appearance quality traits), EQT (eating quality traits), and RQT (reliance quality traits).

Author Response

Reviewer #1

This review paper about metabolomics technology for meat is well organized and written. Considering the recent development of analysis technology and the growing importance of evaluating or predicting meat quality and using these technologies to secure the quality and trust of meat, I think this review paper has a timely meaning. However, one thing I would like to advise for this review paper is that the authors suggested four traits of meat quality decisins, and please refer to Joo et al. (2013, Meat Science 95, 828-836) paper, which divides meat quality into AQT (appearance quality traits), EQT (eating quality traits), and RQT (reliance quality traits).

Reply: Thank you for the positive evaluation and valuable comments. We have reorganized the “3.1. Metabolomics in meat quality” section by dividing meat quality into appearance quality traits (AQT), eating quality traits (EQT) and reliance quality traits (RQT) according to the reviewer’s suggestion. This section has also been rewritten by reviewing metabolomics researches in colour, pH and meat texture of AQT, tenderness and flavour of EQT, and freshness of RQT.

Reviewer 2 Report

The submitted paper provides an overview of the latest methods used in metabolomics technology and gives examples of their use in recent meat science issues. As this is a very current topic, and the paper is well-written, informative and easy to read.  

Some of the following remarks may further improve the text:

General remark:  It would be interesting to hear the authors’ opinion/projection about the feasibility of the metabolomics technique usage in practice, besides scientific studies? E.g. in meat quality control schemes or at slaughter lines?  And what would be the estimated cost of this kind of control?

Row 152: Please add the space after the bracket

Row 452: “omics” instead of “comics”

Row: 485:  as a comment – several cases of substitution of beef with horse meat has been reported recently in the EU    

Row: 497-501: also a comment - a bit strange because lamb is more in demand and has a higher price than mutton in many markets.  

Author Response

Reviewer #2

The submitted paper provides an overview of the latest methods used in metabolomics technology and gives examples of their use in recent meat science issues. As this is a very current topic, and the paper is well-written, informative and easy to read. 

Some of the following remarks may further improve the text:

  1. General remark: It would be interesting to hear the authors’ opinion/projection about the feasibility of the metabolomics technique usage in practice, besides scientific studies? E.g. in meat quality control schemes or at slaughter lines? And what would be the estimated cost of this kind of control?

Reply: Thank you for this valuable comment. As commented by reviewer #3, although extensive researches have been performed in meat science, it is far from practical applications. Most of the studies we cited focus on profiling metabolites and identifying metabolic biomarkers regarding meat quality analysis and authentication. The feasibility of the metabolomics technique usage in practice remains a challenge.

The above issue has been addressed by adding a discussion at the end of the manuscript, which clarifies the limitations and future trends of the practical usage of the metabolomics approach in meat quality analysis and authentication.

  1. Row 152: Please add the space after the bracket

Reply: A space has been added after the bracket.

  1. Row 452: “omics” instead of “comics”

Reply: The “comics” has been corrected to “omics”.

  1. Row: 485: as a comment – several cases of substitution of beef with horse meat has been reported recently in the EU

Reply: We have searched literature related to the substitution of beef with horse meat. An application of targeted metabolomics in the authentication of beef versus horse meat using 60 MHz 1H NMR spectroscopy has been added to the text (Moreover, Jakers et al. developed a 60 MHz 1H NMR based targeted metabolomics to dif-ferentiate between beef and horse meat and concluded that 60 MHz 1H NMR represent a feasible high-throughput approach for screening raw meat [164]).

  1. Row: 497-501: also a comment - a bit strange because lamb is more in demand and has a higher price than mutton in many markets.

Reply: Thank you for this valuable comment. We have checked the literature that we cited. We are sorry that we made a mistake when describing the cited studies. In fact, lamb is more in demand and has a higher price than mutton in many markets. The corresponding text has been corrected as follows: Lamb meat is derived from sheep at the age of no more than 12 months or without permanent incisor teeth, and mutton is defined as meat from sheep of 1–3 years old. Compared with mutton, lamb meat always has a higher demand and retail price in the market [165]. In order to pursue economic interests, illegal producers often use mutton to replace lamb meat. The high-throughput metabolomics approach combined with multivariate data analysis has been proven to distinguish lamb from mutton effectively [166].

Reviewer 3 Report

The article is interesting and original. It summarizes the hitherto achievements in the field of metabolomics in the science of meat. It is worth noting that these are the beginnings of the use of metabolomics in the science of meat. Rather, the results of these studies help to understand the metabolic changes occurring in the meat. However, it is far from practical applications suggested by the title of the article. In the results of the studies cited by the authors, only indications were obtained regarding the differences between the studied groups of animals, without any precise qualitative and quantitative indication, which would enable precise identification of meat defects, types of meat or specific applications. So in my opinion, the title should be more general, for example "Application of Metablomics Technology in Meat Science" or "Current state of metabolomics research in meat science". The more so that the authors themselves emphasize the current difficulties and barriers to overcome in the practical application of Metabolomics.

Author Response

Reviewer #3

The article is interesting and original. It summarizes the hitherto achievements in the field of metabolomics in the science of meat. It is worth noting that these are the beginnings of the use of metabolomics in the science of meat. Rather, the results of these studies help to understand the metabolic changes occurring in the meat. However, it is far from practical applications suggested by the title of the article. In the results of the studies cited by the authors, only indications were obtained regarding the differences between the studied groups of animals, without any precise qualitative and quantitative indication, which would enable precise identification of meat defects, types of meat or specific applications. So in my opinion, the title should be more general, for example "Application of Metablomics Technology in Meat Science" or "Current state of metabolomics research in meat science". The more so that the authors themselves emphasize the current difficulties and barriers to overcome in the practical application of Metabolomics.

Reply: Thank you for this valuable comment. We agreed that metabolomics is far from practical applications and we have changed the title to “Current State of Metabolomics Research in Meat Quality Analysis and Authentication”. It should be noted that we used “……Research in Meat Quality Analysis and Authentication” rather than “……Meat Science” in the title. This is mainly because the scope of “meat science” is too large. In our manuscript, we only reviewed the metabolomics researches in meat quality analysis and authentication. We hope that the revised title “Current State of Metabolomics Research in Meat Quality Analysis and Authentication” could be more suitable for our manuscript.